# New Alphaproteobacteria Thrive in the Depths of the Ocean with Oxygen Gradient

**DOI:** 10.3390/microorganisms10020455

**Published:** 2022-02-16

**Authors:** Miguel Angel Cevallos, Mauro Degli Esposti

**Affiliations:** Center for Genomic Sciences, Universidad Nacional Autónoma de México, Cuernavaca 62210, Morelos, Mexico; mac@ccg.unam.mx

**Keywords:** alphaproteobacteria, bacterial classification, phylogenetic analysis, oxygen, aerobic metabolism, oxygen gradient, evolution of mitochondria

## Abstract

We survey here the Alphaproteobacteria, a large class encompassing physiologically diverse bacteria which are divided in several orders established since 2007. Currently, there is considerable uncertainty regarding the classification of an increasing number of marine metagenome-assembled genomes (MAGs) that remain poorly defined in their taxonomic position within Alphaproteobacteria. The traditional classification of NCBI taxonomy is increasingly complemented by the Genome Taxonomy Database (GTDB), but the two taxonomies differ considerably in the classification of several Alphaproteobacteria, especially from ocean metagenomes. We analyzed the classification of Alphaproteobacteria lineages that are most common in marine environments, using integrated approaches of phylogenomics and functional profiling of metabolic features that define their aerobic metabolism. Using protein markers such as NuoL, the largest membrane subunit of complex I, we have identified new clades of Alphaproteobacteria that are specific to marine niches with steep oxygen gradients (oxycline). These bacteria have relatives among MAGs found in anoxic strata of Lake Tanganyika and together define a lineage that is distinct from either Rhodospirillales or Sneathiellales. We characterized in particular the new ‘oxycline’ clade. Our analysis of Alphaproteobacteria also reveals new clues regarding the ancestry of mitochondria, which likely evolved in oxycline marine environments.

## 1. Introduction

This article is intended to cover diverse topics related to the evolution and classification of Alphaproteobacteria and their adaptation to different ambient oxygen levels, especially in marine environments. Alphaproteobacteria are the second largest and most likely also the second oldest class in the vast *phylum Proteobacteria* [1,2,3,4,5]. Their fundamental classification comprises various orders that have been defined since 2007 [2] but have a very different distribution in nature. The most common order is that of the Rhizobiales, which are widespread in earth environments and have been studied predominantly because of their economic (agricultural) and medical impact [6,7]. The second most common order is that of Rhodobacterales, including purple photosynthetic bacteria that have traditionally dominated the biochemical and genetic studies on Alphaproteobacteria [1,4], plus an ever-increasing number of marine non-photosynthetic bacteria [8,9]. 

*Rhodospirillum*, another purple photosynthetic bacterium, has been considered a type of genus of the order Rhodospirillales, which is metabolically and phylogenetically more diverse than Rhodobacterales [1,4]. Recent studies have proposed an ample revision of the Rhodospirillales because this order is polyphyletic [10,11,12,13]. However, the current classification of the various taxa that are affiliated with or can be associated to the Rhodospirillales remains unsettled in several cases, because the taxonomic ranks in which the order is subdivided in NCBI taxonomy often do not correspond to those introduced in GTDB taxonomy, which splits Rhodospirillales in many orders and additional families [10,12,14].

We summarize in Table 1 the current subdivisions of Rhodospirillales according to NCBI taxonomy, which has adopted the revised classification proposed by Hördt et al., 2020 [13], compared with those of GTDB taxonomy [12,14]—Release 06-RS202 (27 April 2021). The latter has recently added other taxonomic divisions from oceanic metagenomes [14,15] that are broadly associated with Rhodospirillales (Table 2). One reason for the ranking discrepancies shown in Table 1 is that different Rhodospirillales genomes display different evolutionary rates [10,13,16]. For instance, *Tistrella* evolves much slower than members of the Acetobacteraceae family, such as *Roseomonas*. Given this heterogeneous behavior and the evident inconsistencies in the taxonomic rankings of the NCBI and the GTDB taxonomy (Table 1), we propose to define Rhodospirillales as a ‘super-order’. The late Cavalier-Smith suggested this kind of taxonomic division while criticizing the excess of orders and families used by GTDB taxonomy to classify bacterial lineages [16]. Such subdivisions could be retained under the super-order Rhodospirillales, despite the fact that they have been defined solely by phylogenomic analysis [12,14], which contrasts with traditional classifications based upon multiple phenotypic and genotypic features (multiphasic taxonomy) [13,16].

A clear advantage of GTDB taxonomy is that it provides a detailed set of subdivisions to automatically classify many MAGs derived from large assembly studies of marine or other environments [12,14,15,17,18,19]. The limits of this phylogenomics-based classification are not only the incongruencies with the NCBI taxonomy (Table 1), but also the number and type of protein markers that can be assigned to a given MAG to estimate its taxonomic placement [12]. GTDB classification is based upon normalized distances of clades in a reference phylogenetic tree [14] built with a super-matrix alignment of up to 120 marker proteins [12]. Such markers are predominantly ribosomal proteins, but also include different proteins that inevitably have different evolutionary histories, for example cytidine triphosphate synthase (CTP synthase) [10,12]. Concatenation of so many single marker proteins with disparate evolutionary histories does not necessarily improve phylogenetic resolution, leading to ambiguous placements of taxa in phylogenetic trees [20]. Moreover, such trees most likely reflect evolutionary changes of bacterial ribosomes over those occurring in central metabolism [10]. 

**Table 2 microorganisms-10-00455-t002:** Distribution of Alphaproteobacteria lineages in three marine metagenomes. Data represent normalized fractions of the total number of Alphaproteobacteria taxa present in each metagenome (underlined at the bottom) and were elaborated from refs [15,18,19] in the order of the table columns.

Alphaproteobacteria Lineage	Ocean DNA	Saanich Inlet	Black Sea
Clade a: Rickettsiales and TMED109	0.082	0.023	0.048
HIMB59	0.043	0.035	0.016
Holosporales	0.002	0.012	0
Clade b: Rhodospirillales	0.180	0.189	0.317
UBA828—close to Geminicoccales ^1^	0.008	0.035	0
SP197—close to Geminicoccales ^1^	0.008	0.020	0
UBA8366—close to Kiloniellales ^1^	0.008	0.004	0
UBA6615—close to Kiloniellales ^1^	0.008	0.055	0
Other subdivisions Rhodospirillales-related	0.076	0.199	0.095
Unclassified Rhodospirillales	0.003	0.016	0.016
Rhodospirillales super-order ^2^	0.256	0.516	0.429
‘oxycline clade’—GCA_002731375	0.008	0.125	0.016
NEW—UBA2966	0.005	0.054	0
Clade c—SERIK	0.011	0.042	0
Pelagibacterales	0.078	0 ^4^	0.032
Rhizobiales	0.088	0.027	0.032
Rhodobacterales	0.323	0.107	0.175
Caulobacterales	0.072	0	0.048
Sphingomonadales	0.076	0	0
Parvibaculales	0.026	0	0.079
Classified not in other metagenomes	0.032	0.057	0.016
GCA-2701885—MarineAlpha2	0.008	0.004	0
Unclassified and other Alphaproteobacteria	0.066	0.016	0.016
Total Alphaproteobacteria ^3^	9949	256	63

^1^ According to the ML trees [14]. ^2^ Super-order definition as proposed here. ^3^ Total number of taxa equal 1. ^4^ Several MAGs classified as Pelagibacteraceae [18] turned out to be HIMB59 relatives.

We will examine here problems affecting the branching order, and therefore the evolutionary relationships of groups of Alphaproteobacteria irrespective of their precise classification. Such problems became evident after detailed analysis of currently known Alphaproteobacteria, undertaken using an integrated evaluation of phylogenetic trees obtained with single marker proteins, such as the NuoL subunit of complex I (Figure 1), with functional traits of aerobic metabolism [4,21]. New clades of Alphaproteobacteria that predominantly live in oxycline environments have emerged from this integrated analysis and their placement in the phylogeny of Alphaproteobacteria will be discussed in depth. 

## 2. Materials and Methods

To evaluate the phylogeny of the various taxonomic divisions of Alphaproteobacteria we have combined phylogenetic analysis with an in-depth evaluation of metabolic traits [4]. Phylogenetic analysis relied on alignments of large selections of protein sequences (usually over 100) representing taxonomic ranks of interest and with significant amino acid variation [10,21,22,23]. Such alignments were manually implemented after rounds of automated alignment with MUSCLE within the MEGA software (versions 5, 6 and X), as described previously [10,21,22,23]. Only limited trimming of the N and C termini was undertaken to preserve sequence information [23,24], while local sequence matches were iteratively refined using hydropathy analysis and comparison with available 3D-structures [21,22,23,25]. In phylogenetic analysis we routinely used NuoL, the largest membrane subunit of complex I (NADH-quinone oxidoreductase) that has close homologs in the ND5 protein encoded by the mitochondrial DNA of eukaryotes [22,25]. As a single protein marker, NuoL offers several advantages [4,10,22]: it has about 650 residues in bacteria, but requires several inserts for aligning to its mitochondrial homologs, thereby producing alignments exceeding 800 amino acid positions—comparable to 16S rRNA sequences conventionally used for classifying bacteria [2,13]; it is present in aerobic as well as facultatively or obligate anaerobes [23], therefore constituting a nearly ubiquitous bioenergetic trait; NuoL has a strong phylogenetic signal across the whole span of Proteobacterial lineages [3,4,11,22]; it has been regularly included in the sets of proteins used to generate trimmed concatenated alignments for phylogenetic trees of Alphaproteobacteria and mitochondria [3,5,11,26]. Moreover, NuoL is the only subunit of complex I present in poor quality MAGs that have unique phylogenetic placements, such as MarineAlpha1-Bin1 [11]. Other proteins used as single phylogenetic markers were: the NuoD subunit of complex I, corresponding to mitochondrial Nad7/49 kDa binding quinones [11,22,23,26]; COX1 and COX3 of cytochrome *c* oxidase, family A of Heme Copper Oxygen reductases (HCO) or COX [11,21,23,26,27]; cytochrome *b* of complex III (ubiquinol-cytochrome *c* reductase) [4,11,26]; and CTP synthase, a soluble enzyme used previously as a marker [10,12]. Some of these proteins were also concatenated, but only to other subunits of the same enzyme complex, for example COX1, COX and COX3. 

Phylogenetic analysis of single or concatenated proteins was undertaken predominantly with Maximum Likelihood (ML) inference using the program IQ-Tree [28] as previously reported [23]. The most common models for amino acid substitution were WAG, LG and the mixture model EX_EHO. Additional phylogenetic trees were obtained with the NJ approach using MEGA5 and the JTT or BLOSUM62 model [3,23]. Functional analysis of genomes retrieved from NCBI databases (nr—non redundant; genomes and proteins) was undertaken by iterative PSI-BLAST searches of trait-defining proteins, such as those studied previously [10,21,22,27]. Traits, such as COX or other terminal oxidases, were carefully inspected not only for the conservation of the catalytic residues required for their function, but also for the completeness of the gene clusters (operons) in which the catalytic and accessory proteins are present in conserved sequences [21,23,27]. The *ubiTUV* genes that are required for the biosynthesis of ubiquinone (Q) under anaerobic conditions [29] were identified by their conserved domain structure and genomic contiguity. We used the same phylogenomic analysis described above to evaluate the presence of key proteins for the nitrogen (N) cycle [17,27,30]. This applied also to the phylogenetic evaluation of NosZ proteins. 

Consistent (re)annotation of the identified proteins of N metabolism and Q biosynthesis, as well as of unrecognized COX subunits and accessory proteins, was undertaken systematically in an in-house repository of Alphaproteobacteria genomes downloaded from NCBI resources. A script in R was then used to search for the distribution of trait-defining proteins across the various genomes and selections thereof. Bioenergetic proteins of Alphaproteobacteria were compared with sets of their eukaryotic homologs that were predominantly taken from Archaeplastida and related non-photosynthetic protists, such as *Rhodelphils* [31] and *Palpitomonas* [32]. 

## 3. Results and Discussion

### 3.1. A General Survey of the Phylogeny of Alphaproteobacteria

Alphaproteobacteria constitute a large class of phenotypically diverse prokaryotes that has been divided in eight major orders and many families [2,5], predominantly with the conventional analysis of their 16S rRNA and phenotypic properties [1,2,9,13]. To evaluate and graphically render the phylogeny of these taxonomic divisions we have routinely used NuoL, the largest membrane subunit of complex I [22,25], as a single, informative protein marker, for the reasons presented in Materials and Methods. Figure 1 shows a representative ML tree obtained with a very broad set of NuoL proteins from Alphaproteobacteria and Eukaryotes. With few exceptions, notably Pelagibacterales as in previous studies [13], these proteins represent all major orders and families in which Alphaproteobacteria are currently classified, thereby providing a graphical view of the relative abundance and phylogenetic diversity of taxa currently affiliated with the class. The tree is rooted on the distant NuoL homologs of Magnetococcales, generally considered the basal lineage of Alphaproteobacteria [2,4,5,13], and shows all the other bacterial lineages in a single clade that is sister to that of the mitochondrial ND5 homologs.

The tree in Figure 1 shows an ‘Alphaproteobacteria-sister’ to mitochondria topology that is equivalent to that initially reported using 24 concatenated proteins, including NuoL and many marine MAGs [11]. The same clustering has been recently reproduced with different taxonomic samplings of either Alphaproteobacteria or mitochondria [26,33]. However, the most common topology of phylogenetic trees combining Alphaproteobacterial and mitochondrial proteins shows the mitochondrial clade nested within the early branching part of the trees, often in a sister position to the Rickettsiales order [2,4,11,26]. Leaving aside the interpretation of these different topologies—see [11,24,26] for a detailed discussion of the topic—the tree in Figure 1 shows a fundamental feature relevant to the overall phylogeny of Alphaproteobacteria that has been scarcely noted before. The branching order of the various lineages follows a sequence of three major clades, which are labeled **a**, **b** and **c** here. Clade **a** includes the order of Rickettsiales and appears to be closely related to MAGs that have similarly low G+C content in their genomes, such as those affiliated to the TMED109 order of GTDB taxonomy [11,14]. Clade **b** includes most Rhodospirillales and is generally separated from clade **a** by the deep branching family [13] or order [12] of Geminicoccaceae/ales (Table 1). Then, a series of minor orders of predominantly marine taxa, such as Sneathiellales, separates clade **b** from clade **c**, which includes all major lineages of Alphaproteobacteria, from Rhizobiales to Rhodobacterales (Figure 1). An equivalent three-clades sequence can be discerned in the branching order of phylogenetic trees reported earlier, for example in studies on Alphaproteobacteria only [3,5,13], as well as in studies focusing on the Alphaproteobacterial ancestry of mitochondria [11,26,33]. Indeed, clade **c** fundamentally corresponds to the ‘Core alphaproteobacteria’ group reported by Martijn et al., 2018 [11].

The branching order of clade **b** and **c** differs from that of Figure 1 in some recent studies, in which these major branches appear as sister clades [5,11,26,33]. The probable reason for this difference is the limited taxonomic sampling of Alphaproteobacteria in those studies, since phylogenetic trees reconstructed with much larger samplings [13,14] do show clade **b** branching before clade **c** as in Figure 1. ML trees reconstructed with other single marker proteins, such as COX3 and NuoD, show a similar branching pattern to that in Figure 1 too (see later Section 3.5). Moreover, the lineages intermixed between clade **b** and **c** (Figure 1) form a constant feature in phylogenetically broad trees including all such lineages [13,14]. Only Sneathiellales have been regularly included in other studies [5,11,26], while single representatives of Kordiimonadales and Iodidimonadales have been shown to cluster with Sphingomonadales in clade **c** [11]. This clustering most likely derives from limited taxonomic samplings of the lineages forming the clade, a common distortion in phylogenic trees [10,13,20]. Here we show that Sneathiellales, Kordiimonadales and other lineages at the base of clade **c** present metabolic features that are essentially intermediate between those of representatives of clade **b** and clade **c**, sustaining the concept that Alphaproteobacteria have progressively evolved from anaerobic or facultatively anaerobic ancestral taxa of Rhodospirillales [4,10] to strictly aerobic taxa of marine Caulobacterales and Rhodobacterales [4,13,21]. The deep divide between clade **b** and **c** in phylogenetic trees, therefore, may derive from a major evolutionary event related to the consolidation of permanent levels of oxygen in primordial earth [21,23], which enabled novel aerobic traits to arise in bacteria [4,21]. Traits, such as methanotrophy and aerobic nitrification, arose first in ancestors of extant Delta- and Gamma-proteobacteria and subsequently permeated the genome of Alphaproteobacteria by waves of Lateral Gene Transfer (LGT), presumably after some Rhodospirillales had already diverged [27]. We introduce here the SERIK group to further corroborate this hypothetical evolutionary scenario, which derives from the phylogeny of Alphaproteobacteria (Figure 1) combined with other considerations [4,10,21,27].

### 3.2. The SERIK Group and New Marine Clades of Alphaproteobacteria

SERIK is an acronym derived from the initial letters of Sneathiellales, Emcibacterales, Rhodothalassiales, Iodidimonadales and Kordiimonadales, following their branching sequence in descending phylogenetic trees, such as that presented in Figure 1. These orders include almost exclusively marine taxa that have been introduced among Alphaproteobacteria in the last decade or so [34,35,36,37]. They have been validated as separate orders, clustering together in a single branch often labeled EKRS (i.e., in alphabetical order), by the recent re-classification of Alphaproteobacteria [13], which has been adopted almost entirely in the current NCBI taxonomy system (Table 1). However, only Sneathiellales is considered a separate order in the GTDB system, in agreement with [5], classifying Emcibacterales, Rhodothalassiales, Iodidimonadales and Kordiimonadales in three families of the large order of Sphingomonadales (Table 1). One of such families, the Rhodothalassiaceae, has been introduced earlier [35], but is considered part of the separate order of Rhodothalassiales in other classifications [13]. In GTDB taxonomy, the family Rhodothalassiaceae also includes the order Iodidimonadales (https://gtdb.ecogenomic.org/searches?s=gt&q=f__Rhodothalassiaceae—accessed on 11 January 2021). Alphaproteobacterium Q-1, which is a consolidated member of the Iodidimonadales [34,36], is photosynthetic as *Rhodothalassium* [35]. However, the photosynthesis trait is absent in all other taxa of the SERIK group; therefore, it likely derives from sporadic LGT of the genetic island carrying all the genes required for photosynthesis, which is frequently exchanged in marine Rhodobacterales [8,9]. 

The different classification of SERIK lineages in NCBI and GTDB taxonomies (Table 1) prompted a detailed phylogenomic analysis of uncultured bacteria that could be associated with SERIK taxa. The search focused on MAGs derived from recent metagenomes of marine environments (Table 2) and was integrated with the phylogenetic analysis of major subdivisions of GTDB taxonomy that cluster together with the Sneathiellales in ML trees [14]. In such trees, the branch containing the Sneathiellales includes the clade of Minwuiales, another minor marine order reported to be distinct from other bacteria [37], plus two subdivisions of marine MAGs considered as separate orders (Table 1): UBA2966, with the representative Rhodospirillaceae ARS1032, and GCA_002731375, named after the genome assembly code of Rhodospirillaceae NP113. Both these MAGs were found by the Tara Oceans Consortium [38] and are represented in the MarineOcean DNA survey [15], as well as in the metagenome of the Saanich Inlet [18], a seasonally anoxic fjord that has been studied as a model for marine oxygen minimum zones (OMZ) [17] (Table 2). 

Although representatives of these orders are not present in the metagenomes of many freshwater environments [39], related taxa have been found in the anoxic depths of Lake Tanganyika, a unique stratified environment of deep freshwater [40] (Figure 2a). The NuoL tree of Figure 1 includes a representative of Lake Tanganyika metagenome and three MAGs classified in the UBA2966 order, which form a distinct branch lying between clade **b** and the SERIK group in NuoL trees (Figure 2a). Seven MAGs from Lake Tanganyika have been classified as Sneathiellales using GTDB [40], but Sneathiellales are marine-specific bacteria [13,41]. Considering their phylogenetic position and other features described below, such taxa of Lake Tanganyika are proposed to be part of novel clades of Alphaproteobacteria that may be characteristic of aquatic environments with steep oxygen gradients; they are cumulatively called new clades here. Alphaproteobacteria MarineAlpha1_Bin1 appears to constitute a precedent for such new clades, since it was reported to lie basal to the equivalent of clade **c** in several phylogenetic trees [11,42] and its NuoL protein clusters with the oxycline clade overlapping o_GCA_002731375 (Figure 2a). The genome of MarineAlpha1_Bin1 is of very low quality and has only a fraction of the set of ‘alphamitoCOGs’ used by Ettema and coworkers to construct phylogenetic trees [11,42]. Yet, such trees invariably positioned this MAG away from other marine MAGs, except for MarineAlpha2_Bin1, which belongs to another GTDB order, o_GCA-2701885, annotated in Table 2. Bioenergetic proteins of this order do not segregate with the same oxycline clade as those of MarineAlpha1_Bin, but close to members of the UBA2966 clade and one MAG from Lake Tanganyika (Appendix A). 

### 3.3. The New Marine Clades Appear to Cluster with Sneathiellales but Have Different Functional Traits

Given their original classification, the Alphaproteobacterial taxa that form part of the new clades introduced here could belong to either the Rhodospirillales ‘super-order’ or the Sneathiellales order. To discriminate between these possibilities, we produced alignments of single marker proteins from representatives of most subdivisions of Rhodospirillales in GTDB taxonomy (Table 2) combined with those of selected taxa of the new clades and of Sneathiellales [14]. Figure 2a shows a typical tree obtained with the NuoL protein indicating that MAGs of the new clades cluster with representatives of either the UBA2966 or the GCA_002731375 order of GTDB taxonomy, away from taxa classified in NCBI taxonomy but within the large clade that includes all cultivated Sneathiellales, such as *Oceanibacterium* and Minwuiales (Figure 2a). Proteins of Lake Tanganyika taxa classified as Sneathiellales [40] form the earliest branches of the clade, which lies in a sister position to the clade, including most proteins from Rhodospirillales (Figure 2a). Although the bootstrap support for the separation of these large clades is not very high, the tree topology in Figure 2a has been reproduced in multiple trees with different models and taxonomic samplings. Since the statistical weight of consistent bifurcating topologies obtained in separate trees is much stronger than branch bootstrap support [20], the separation of Sneathiellales from Rhodospirillales likely reflects a fundamental split in the phylogeny of Alphaproteobacteria, as indicated by much broader trees (see Figure 1 and previous discussion at the end of Section 3.1). Given the clustering of the members of the new clades with Sneathiellales (Figure 2a), it appears that such clades may represent families or sub-orders related to the Sneathiellales order. Consequently, their original denomination of ‘Rhodospirillaceae bacterium’ [18] should be rectified.

Nevertheless, subsequent phylogenetic (Figure 3) and functional analysis (Figure 2b) suggest that the new clades should be considered taxonomically separate from Sneathiellales. The results of our detailed analysis of functional traits linked to aerobic metabolism, summarized in Figure 2b, strongly sustains this possibility. The first piece of evidence regards Q and the reoxidation of its reduced form, ubiquinol, catalyzed by ubiquinol-cytochrome *c* reductase (complex III), which is present in all the taxa in Figure 2b and the great majority of their relatives. Some Sneathiellales can additionally oxidize ubiquinol using low concentrations of oxygen because they possess one or more cytochrome *bd* ubiquinol oxidases of the CIO type [21,27]—top left of Figure 2b. However, members of the new clades and associated MAGs of Lake Tanganyika do not have such alternative oxidases to discharge reduced Q; yet they have the capacity of synthesizing Q also under anoxic conditions, since their genome often contains the *ubiTUV* cluster that is required for this anaerobic biosynthesis [29] (Figure 2b). Genes for ubiTUV proteins are present in Magnetococcales and in a few Rhodospirillales [29], but otherwise are rare in marine Alphaproteobacteria, as verified by detailed genomic analysis. Therefore, their presence in members of the new clades (Figure 2b) indicates an uncommon adaptive trait to anoxic conditions and implies a redox overload on *c*-type cytochromes by ubiquinol-cytochrome *c* reductase. The second piece of evidence regards the discharge of this redox overload, which is carried out by an uncommon variety of terminal oxidases of the HCO superfamily (Figure 2b, cf. [21]). 

All families of HCO are present in Alphaproteobacteria, but infrequently together in the same genome [21,27]. Soil Rhizobiales of the *Bradyrhizobium* genus are an exception, since they have genes for members of HCO family A, B and C in their genome, often with different subtypes of A family COX, such as *ccoNOPQ* [43], corresponding to subtype a-III [21,27]. Although hard evidence is lacking, subtype a-III is likely to operate at lower oxygen concentrations than the main subtype b COX [21,27,43]. Among marine Alphaproteobacteria, COX operon subtype a-III is present in clades of Pelagibacterales living near or within OMZ [44] and a few Rhodospirillales, for example *Inquilinus* [21]. Remarkably, they are concentrated in the MAGs of the new clades, together with C family oxidases and also a specific subtype of B family oxidases, previously labeled ba3-a1 [4,21] (Figure 2b). Hence, the genomes of the new clades of Alphaproteobacteria introduced here have an exceptional combination of three families of HCO terminal oxidases, which clearly distinguishes them from Sneathiellales and other SERIK members. 

Before describing further phylogenetic analysis, it is worth discussing the N cycle [30], another functional trait that is characteristic of the new clades (Figure 2b). Recently, a novel element of this cycle has been identified in the HCO subtype ba3-a1 [21], which has been found to function as a nitric oxide reductase using reduced cytochrome *c* as electron donor [45], similarly to the classical nitric oxide reductase of the N cycle, NOR [17,30,46,47]. The ba3-a1 with NOR-like activity is widespread among members of both the oxycline and NEW—UBA2966 clades (Figure 2b), while only *Minwuia* has classical NOR among the taxa shown in Figure 2b. Several taxa have other enzymes of the N cycle, including Nitrous Oxide (N_2_O) Reductase, usually labeled NosZ after the *nosZ* gene of its catalytic protein [30] (Figure 2b). However, only one representative of each clade is potentially capable of undertaking complete aerobic denitrification, similarly to *Minwuia*: Alphaproteobacteria BS150m-G9 of the oxycline clade, as well as Alphaproteobacteria SIO54-bin61 of NEW—UBA2966. These two MAGs have the cd1 type of nitrite reductase as their relatives and would use ba3-a1 as a NOR to provide the substrate for their NosZ, which clusters in a different clade than that of *Minwuia* in phylogenetic trees (Appendix A). However, the majority of taxa forming the new clades introduced here do not have a functional NosZ and therefore would contribute to N_2_O accumulation in their marine environment [17,46,48]. 

**Figure 2 microorganisms-10-00455-f002:**
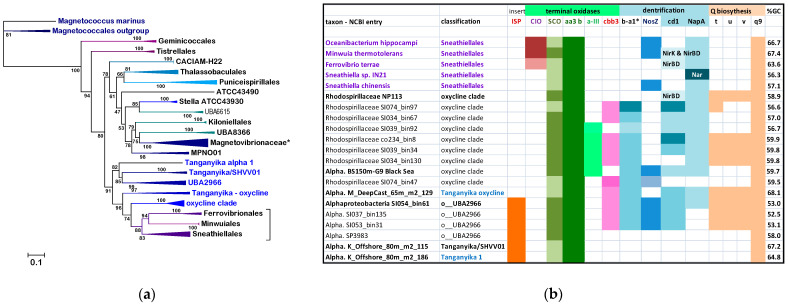
Phylogeny and functional profile of new clades of Alphaproteobacteria. (**a**) The phylogenetic ML tree was reconstructed with the LG model using NuoL proteins from Rhodospirillales and Sneathiellales taxa labeled according to GTDB taxonomy and the clades introduced here (Table 1; * Magnetovibrionaceae include also f_Casp-alpha; the bracket embraces all taxa of Sneathiellales); (**b**) metabolic traits and genomic properties of Sneathiellales and representatives of the clades introduced here. ISP stands for the Iron Sulfur Protein subunit of ubiquinol-cytochrome *c* reductase; the orange boxes indicate the presence of the insert typical of Rhodospirillales that separates the conserved stretches with the Fe-S ligands [27]. SCO (Synthesis of Cytochrome C Oxidase) is a COX accessory protein involved in copper insertion which is often associated with B family HCO [4]. Terminal oxidases include CIO (Cyanide Insensitive Oxidase) of the cytochrome *bd* ubiquinol oxidases [27], which is present only in some Sneathiellales, and one or two HCO A family subtypes, aa3 b and a-III [4,21]. Cbb3 identifies the complete operon of C family HCO [21]. *Subtype ba3-a1 of B family HCO [4] was suspected to be involved in nitrogen metabolism (see [21] and references therein) and has been recently reported to exhibit NO reductase activity, hence re-named eNOR [45]. It is thus part of the N cycle, which is usually incomplete in marine biomes because of the limited distribution of classical NOR (NO reductases) and NosZ (N_2_O reductases) [17,30]. Clade I, TAT-dependent nitrous oxide reductase [17], is annotated as NosZ, while only *Minwuia* has a classical cytochrome *c*-reducing NOR completing aerobic denitrification [30]. See Appendix A for the taxonomic distribution of NosZ. Nitrite (NO_2_) reductases are defined according to their structure (cd1) or gene (NirK for the Cu-dependent and NirBD for the NAD(P)H-dependent) [30]. Except one *Sneathiella*, which has transmembrane Nar, all taxa shown have the NapA type of periplasmic nitrate (NO_3_) reductase [30]. The boxes are colored with increasing intensity when more than one gene/operon is present in the genome. The proteins for Q biosynthesis reported on the right are annotated according to their genes [29]: t for sterol-carrier-like *ubiT*, u and v for *ubiU* and *ubiV* that have a U32 peptidase fold; q9 for CoQ9, an accessory protein shared by Alphaproteobacteria and mitochondria [4]. The last column on the right (%GC) lists the percentage of GC bases in the genomes of the taxa, a parameter strongly correlated with taxonomic relatedness in most Alphaproteobacteria [13]. Here it is shown to vary considerably among Sneathiellales genera, while remaining close to the average value of 59% for the MAGs of the oxycline clade. Note that the classification of Lake Tanganyika MAGs is very tentative.

### 3.4. Functional and Phylogenetic Diversity of the Oxycline Clade

The analysis of the traits presented in Figure 2b suggests that two of the clades of marine Alphaproteobacteria introduced here, namely the NEW—UBA2966 and the ‘oxycline clade’ (Table 2), clearly differ from members of the Sneathiellales in the following functional characters: (1), the absence of ubiquinol oxidases; (2), the capacity of Q biosynthesis under anoxic conditions; (3), the rare combination of HCO terminal oxidases; (4), the presence of ba3-a1 oxidase that may function as NOR; and (5), the abundance of enzymes for aerobic denitrification. These functional diversities suggest a strong divergence from Sneathiellales, despite the clustering observed in trees with limited taxonomic sampling (Figure 2a). We then focused our analysis on the ‘oxycline clade’ because the distribution of its distinctive functional traits is more compact than in the NEW—UBA2966 clade, some taxa of which do not have such traits (Figure 2b). Moreover, proteins from taxa of Lake Tanganyika often intermix with those of NEW—UBA2966; therefore, this clade may have a higher taxonomic rank than the ‘oxycline clade’. We tentatively consider the latter clade as a family level taxonomic division encompassing the previously ill-defined GCA_002731375 order [14] (see Appendix A for further details). 

We used CTP synthase as a single protein marker to expand the phylogenetic analysis of the oxycline clade. This large soluble enzyme is part of the 120 protein markers used for GTDB taxonomy [12] and reacts with nucleotides as the majority of the same proteins. In the absence of sequence data for the 16S RNA in MAGs [17,18,19,38], CTP synthase can provide a valuable marker for taxonomic classification of Alphaproteobacteria, having features complementary with those of NuoL [10]. Figure 3 shows a representative ML tree of this protein from Alphaproteobacterial taxa spanning deep branching Rhodospirillales to early branching Rhizobiales, therefore encompassing the phylogenetic space of the new clades (Figure 1). The oxycline clade is separate from the large clade of Sneathiellales, branching after the clade of Zavarziniales—a lineage originally associated with Acetobacteraceae, but later found to form a separate family [13] or order [12] (Table 1). The NEW—UBA2966 clade lies intermediate between the Zavarziniales and the oxycline clade and together with the latter forms a sister branch to that, including the ERIK lineages (Figure 3). This observation has been confirmed in trees obtained with much wider taxonomic sampling.

**Figure 3 microorganisms-10-00455-f003:**
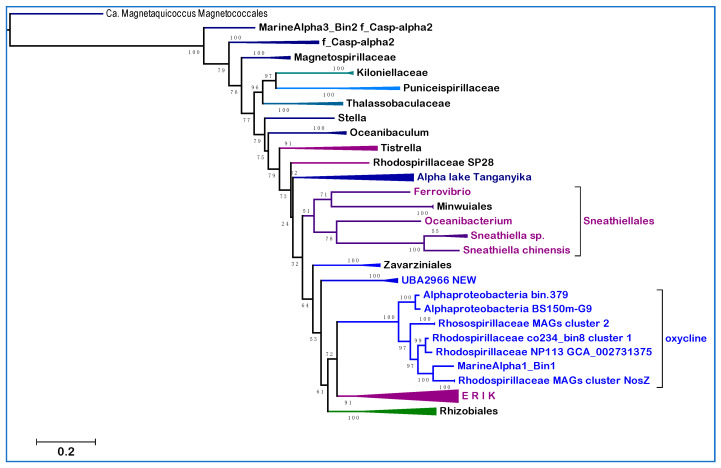
Phylogeny of the oxycline clade using the CTP synthase marker. The ML tree was reconstructed with the LG model using the sequences of CTP synthase [10] from an assortment of Rhodospirillales and Sneathiellales taxa similar to that used for the tree in Figure 2a, plus various representatives of the ERIK orders of the SERIK group and some early branching Rhizobiales, for example *Parvibaculum*. The alignment included 60 sequences with 568 amino-acid sites. The labelling of the various taxa of the oxycline clade follows the nomenclature shown in Appendix A.

The tree in Figure 3 additionally shows a granular definition of subdivisions within the oxycline clade, which tentatively include three major species-level clusters as detailed in Appendix A. Notably, the CTP synthase of MarineAlpha1_Bin1 associates with full support to the cluster defined as NosZ for the presence of this trait in its MAGs (Appendix A). Hence, MarineAlpha1_Bin1 can be considered a *bona fide* member of the oxycline clade. Intriguingly, all the CTP synthase sequences of the oxycline clade show an insert of 10–12 residues that appear to be unique among Alphaproteobacteria, thereby constituting a molecular signature useful for classification.

Next, we enlarged the taxonomic breadth of our analysis, encompassing all known lineages of Alphaproteobacteria using different protein markers. Although the branching sequence of the oxycline clade and associated new clades was essentially confirmed, we noted their tendency to segregate with Minwuiales and *Ferrovibrio*, while the core group of Sneathiellales branched in apparent sister position to the order Sphingomonadales, as reported earlier [5,11,26,42]. To verify whether Sphingomonadales may be functionally related to Sneathiellales, we have analyzed the lineage-specific frequency of the gene encoding NosZ, the copper enzyme that completes the denitrification pathway [30] (Figure 4). Results obtained with two alternative approaches for evaluating the lineage-specific frequency of this gene show that Sphingomonadales are clearly distinct from Sneathiellales and other SERIK lineages, some of which, in particular Iodidimonadales, display the highest frequency (Figure 4). 

### 3.5. The Expanded Phylogeny of Alphaproteobacteria and Its Problems

As single phylogenetic markers, NuoL and CTP synthase suffer from compositional heterogeneity problems similar to those experienced when these proteins are concatenated with other markers in super-matrix alignments [3,5,11,12,26]. It is generally considered that the major source of such problems is the strong bias towards GC-poor, and consequently AT-rich genomes in lineages with an endoparasitic lifestyle, such as Rickettsiales [5,26] and marine taxa with streamlined genomes: MarineAlpha9 [11], HIMB59 [3,5,11], and Pelagibacterales [3,11,44]. The genome-derived compositional bias of coded proteins, combined with their high mutation rate [5,11,26], produce phenomena of Long Branch Attraction (LBA) [3,4,5] between AT-rich Alphaproteobacteria, and also between these bacteria and similarly AT-rich mitochondria [3,4,5,11,24,26,33]. In the ML tree of Figure 1, LBA has influenced the clustering of MarineAlpha9 and Holosporales within clade **a**. Holosporales are either a separate order [13,14] or family [5,10] of obligate endocellular parasites that have been traditionally associated with Rickettsiales (see [5] and References therein). However, ML trees reconstructed from NuoL alignments of smaller or different taxonomic samplings than that used in Figure 1 do separate Holosporales from clade **a**, clustering them within clade **b** as it is now considered to be correct [5,26,33]. Nevertheless, MarineAlpha9, HIMB59, and often Pelagibacterales, remain clustered with Rickettsiales in most NuoL trees, suggesting underlying LBA effects, most likely due to the bias towards hydrophobic amino acids in AT-rich genomes [5,26], which would inevitably affect a strongly hydrophobic membrane protein such as NuoL.

After analyzing various proteins shared by Alphaproteobacteria and mitochondria, we found that most LBA effects and compositional heterogeneity problems disappear in phylogenetic trees reconstructed from taxonomically broad alignments of the NuoD subunit of complex I (Figure 5). Bacterial NuoD is a close homolog of Nad7 encoded by the mitochondrial DNA of Archaeplastida and other protists [22,31,32], but is encoded in the nuclear DNA of other eukaryotes and labeled 49kDa subunit [22,25]. NuoD is a soluble protein and therefore is expected to be less affected by genome-induced compositional bias than transmembrane proteins such as NuoL. Moreover, it is very conserved, especially at the C-terminus where vestigial Cys ligands of the Ni-Fe cofactor that were present in its ancestor, the large catalytic subunit of hydrogenases, are sometimes retained [22]. We found this to be the case in the NuoD sequences of MarineAlpha9 taxa and a few other marine Alphaproteobacteria that cluster with them (for example Rhodospirillaceae SP28, Figure 5), consequently anchoring such taxa at the basis of Alphaproteobacterial phylogeny. Indeed, vestigial Cys ligands in NuoD were previously found only among deep branching bacteria [22].

Detailed examination of the NuoD tree in Figure 5 indicates that the proteins from AT-rich lineages segregate away from Rickettsiales, which remain the fundamental component of clade **a**. Holosporales cluster in the middle of clade **b** as in trees obtained after complex correction of concatenated alignments for compositional heterogeneity [5]. HIMB59 and related taxa such as MarineAlpha5 cluster with another branch of clade **b**, while Pelagibacterales cluster within clade **c**, just before the separation of Rhizobiales and other major lineages of the class (Figure 5). These branching positions are the same as those obtained after stationary-trimming of multiple protein alignments to sequentially remove site heterogeneity and reduce compositional bias [11]. The branching order of Pelagibacterales is also equivalent to that reported in Bayesian or ML trees corrected with different procedures to limit LBA effects [3,26,33]. Here, using a suitable single marker protein and wide taxonomic samplings, we have obtained phylogenetic trees that show the most appropriate segregation of different AT-rich genomes, as if we had applied sophisticated procedures to minimize LBA and compositional bias artefacts. Such procedures have been criticized because they inevitably reduce phylogenetic signal [24], removing sites without considering their position in the structural architecture of the aligned proteins. Ultimately, this work shows that it is not necessary to apply sophisticated bioinformatic corrections for obtaining a comprehensive phylogenetic picture of all major lineages of Alphaproteobacteria. Therefore, that ‘the only way to counter artefacts [of this kind] is by removing compositionally biased sites, or taxa’ [5] appears to be a bioinformatic myth. Conversely, the relatively short length and strong conservation of the NuoD protein (the alignment used for the tree in Figure 5 has less than 500 amino acid positions) prevents discrete resolution of several branches of clade **c**. These are undoubtedly better resolved in NuoL trees (Figure 1), or with concatenated protein alignments (cf. [11,26]). 

To reduce the problem of limited resolution in the branches of clade **c**, the alignment used in Figure 5 was extended to NuoD proteins of late branching lineages, such as Hyphomonadaceae and *Robiginitomaculum*, which are now considered to be sisters rather than part of the related orders of Rhodobacterales and Caulobacterales [13,49]. The MAGs classified in the TMED25 division of Rhizobiales [14], which are abundant in marine metagenomes (Table 2 and [15]), cluster together with Hyphomonadaceae and Robiginitomaculaceae (Figure 5), as expected from the highly derived genomes of these taxa. On the other hand, Parvibaculaceae are shown to be the earliest branching family of Rhizobiales (Figure 5), consistent with some studies [6,13], but apparently in contrast with another [7].

### 3.6. Phylogenetic Trees of NuoD with Both Alphaproteobacteria and Mitochondria

Given the proven value of NuoD as a single phylogenetic marker for the phylogeny of Alphaproteobacteria (Figure 5), it was natural to extend its alignment to homologous Nad7 proteins encoded by the mitochondrial DNA of protists [22]. Representative ML trees, as that shown in Figure 6, displayed the Alphaproteobacteria-sister topology to the mitochondrial clade as in Figure 1 and previously reported trees [11,26,33]. The tree in Figure 6 is presented in ascending order for enabling its direct comparison with such published trees [11,26,42]. It shows a decreased resolution of clade **c** branches with respect to the tree obtained with Alphaproteobacterial proteins only (Figure 5), also because of its smaller taxonomic sampling. Moreover, some LBA effects, presumably due to the inclusion of mitochondrial sequences, seem to have pulled the branch of the HIMB59 order in between clade **a** and **b** (Figure 5 and Figure 6). However, the separate tree position of major AT-rich lineages, such as Pelagibacterales, remains the same as in Figure 5, without any clustering of Rickettsiales with mitochondria (Figure 6). By contrast, clustering of mitochondria with Rickettsiales and Holosporales was regularly found in phylogenetic trees constructed with concatenated alignments of COX1-2-3 proteins, or single accessory proteins such as CtaG_Cox11 [23]. Similar findings have been reported using concatenations of these and other bioenergetic proteins [24]. 

Intriguingly, the addition of NuoD sequences from the early branching group of MarineProteo1 [11] produces a new branch subtending the sister clades of mitochondria and all Alphaproteobacteria in NuoD trees (Figure 6), as well as in NuoL trees. These results fundamentally agree with the increasing number of studies [11,26,33,42] supporting the hypothesis that the progenitors of mitochondria did not emerge from within the Alphaproteobacteria class, but from a related lineage of Proteobacteria [11,26]. The hypothesis derives from a strict interpretation of phylogenetic trees as that in Figure 6 [11,26], assuming that such trees can capture the evolutionary information retained in the proteins of Alphaproteobacteria and mitochondria that are available today. However, this assumption is not guaranteed a priori, since the phylogenetic signal of an ancient transition, such as mitochondrial evolution, estimated to be around 1.8 billion years old [21,33], might have been lost almost entirely in contemporary proteins [4,20,23,26]. Combining together more and more proteins of different type and evolutionary history to reconstruct phylogenetic trees [11,26] does not necessarily enhance the residual phylogenetic signals that some of these proteins may have, producing instead additional sequence entropy that is difficult to disentangle from genuine molecular information of evolutionary value [16,20]. This can be obtained relatively easily by focusing on protein markers with a solid phylogenetic signal [20,50], such as NuoD (Figure 5 and Figure 6). Indeed, the classical single marker 16S RNA still produces valid phylogenetic information for the classification of Alphaproteobacteria, provided that exhaustive taxonomic samplings are used to reconstruct its trees [13].

## 4. Conclusions

We have discussed here various aspects of the phylogeny of Alphaproteobacteria that may be of interest to a variety of scientists working in areas of microbiology, microbial ecology, oceanography, bioenergetics and molecular evolution. Perhaps the most valuable aspect with potential practical applications is the critical evaluation of the NCBI and GTDB taxonomy that are routinely employed, often with automatic bioinformatic tools [14,51], to provide a broad classification of MAGs retrieved from marine and other environments. Biologists and taxonomists are not bioinformatic experts [51,52,53,54], while bioinformatic experts traditionally overlook biological issues [16,52]. Yet, only the integration of different expertise can deal with the increasing demand of classifying non-cultured bacteria derived from metagenomic studies [53,54]. Perhaps one day the differences and incongruencies that currently exist in two major taxonomies (Table 1) and other aspects of bacterial classification [51,52,53,54] will be resolved. In the meantime, experimental studies should ideally document both taxonomic classifications for the MAGs they report, as in Ref. [40], but considering the caveats discussed below. Incorrect or partial classification of bacteria is not a trivial problem in microbial ecology, given the focus on functional diversity in natural and modified ecosystems [17,18,40,43,51,52]. Our experience indicates that GTDB taxonomy probably encompasses most lineages found to date in marine environments, but is yet inadequate to classify all MAGs from aquatic biomes, for example the anoxic strata of Lake Tanganyika [40]. Moreover, current GTDB taxonomy fails to properly classify MAGs that belong to the SERIK group, presumably because of excessive clustering with the Sphingomonadales (Table 1 cf. [12]). Our phylogenetic and functional analyses indicate that Sphingomonadales should be considered taxonomically separate from the SERIK group (Figure 1, Figure 4 and Figure 5), contrary to recently published trees [11,26]. Given the abundance of SERIK and Sphingomonadales taxa in some marine biomes (Table 2), this distinction may have a considerable impact on the classification of marine MAGs. For instance, several taxa classified as ‘Kordiimonadaceae bacterium’ in the metagenome of Saanich Inlet [18] actually belong to the Emcibacterales order. Conversely, a significant proportion of the over 300 MAGs currently listed in NCBI taxonomy under the umbrella name of ‘Rhodospirillaceae bacterium’ belong to other families of the Rhodospirillales ‘super-order’ (Table 1), or cluster with the new clades presented in this article. 

We have introduced here new clades of marine Alphaproteobacteria that have potential relatives among MAGs of Lake Tanganyika (Table 2, Figure 2 and Appendix A). One clade is named NEW—UBA2966 (Table 2) because it overlaps the order UBA2966 [14] that is currently present in GTDB with four MAGs (https://gtdb.ecogenomic.org/searches?s=al&q=o__UBA2966, accessed on 21 January 2022). We found 12 additional MAGs that may belong to the same order in the metagenome of Saanich Inlet [18] and about 50 in the OceanDNA repository [15]. The other clade, provisionally named ‘oxycline’, is better resolved phylogenetically (Figure 3) and functionally (Figure 2b) than the other new clades and overlaps with the order GCA_002731375. This taxonomic unit is most likely over-ranked, since its two taxa represented in GTDB [14] closely cluster with those found in the Saanich Inlet [18] and other marine environments [19,38] (Figure 3 and Appendix A). About 80 MAGs in the OceanDNA repository [15] appear to be members of the same clade and in the majority of cases (74%) have been recovered from OMZ or equivalent niches with very low oxygen concentrations—probably the highest percentage for GTDB-classified marine MAGs [15]. For this reason, we have introduced the name ‘oxycline’ to classify a group of Alphaproteobacteria MAGs that predominantly thrive in marine niches with low or gradient oxygen levels and cluster between clade **b** and **c** in phylogenetic trees (Figure 1, Figure 3 and Figure 5; Appendix A). The oxycline clade is probably a family containing at least three species-level clusters (Appendix A). One of such clusters includes MarineAlpha1_Bin1 (Figure 3), which was retrieved from a deep zone in the middle of the Atlantic Ocean [11] apparently not associated with known OMZ. However, the oxycline clade is clearly distinct from the GTDB order GCA-2701885 that includes MarineAlpha2 [11], all representatives of which live in marine environments with normal oxygen levels [15]. 

An interesting aspect that has emerged from the functional profiling of the new clades of Alphaproteobacteria introduced here (Figure 2b) is that detailed functional analysis defines evolutionary separations that phylogenetic trees may hardly resolve [52]. Hence, the information that even the most sophisticated phylogenetic tree generates on deep branches and apparently strongly supported clades should be interpreted with caution [20,50], particularly when data for key functional traits are scant, as often is the case when using MAGs [52]. This conclusion bears relevance also to the evolutionary relationships between Alphaproteobacteria and mitochondria, since recent studies on mitochondrial origin have relied on MAGs and an increasing number of concatenated proteins of different kinds [11,26,42], with diverse evolutionary histories and phylogenetic signal that may confound deep branching bifurcations in phylogenetic trees [20,50]. Moreover, many of the new MAGs introduced in these studies have been poorly characterized in terms of functional traits related to mitochondrial physiology [10,24]. In sum, an accurate taxonomy of Alphaproteobacteria is critically required also for studying the evolution of mitochondria. It is hoped that the present article will help advancing current and future efforts to address this major issue in evolutionary biology.

## Figures and Tables

**Figure 1 microorganisms-10-00455-f001:**
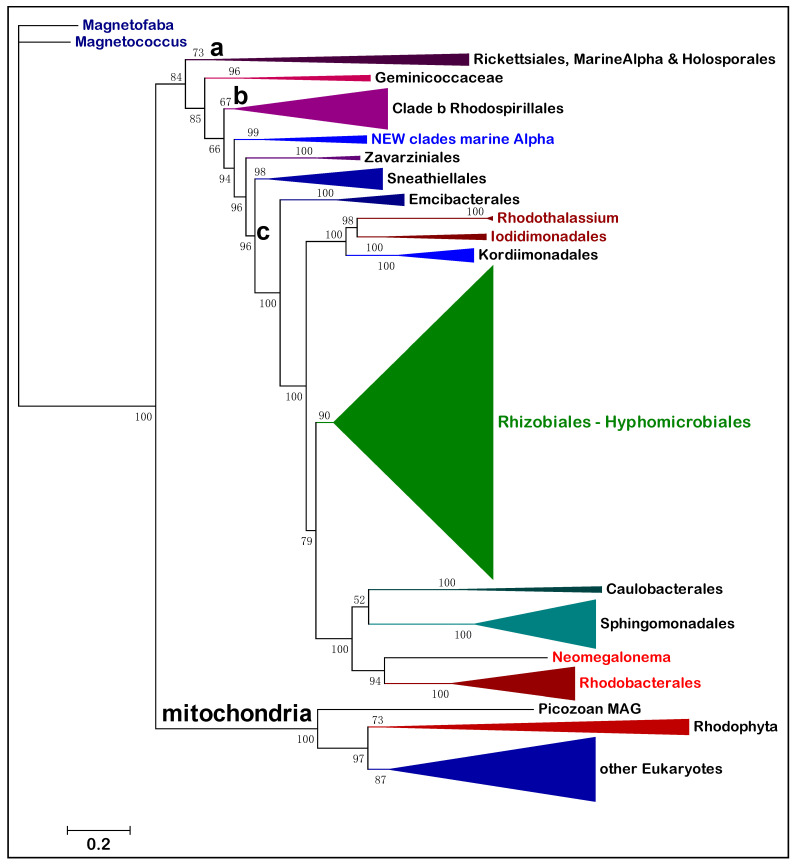
Phylogenetic ML tree of Alphaproteobacteria vs. mitochondria using the NuoL subunit of complex I. The alignment included 242 bacterial NuoL and 64 ND5 homolog proteins from mitochondria, for a total of 818 amino acid sites. The expanded set of proteins is shown in Appendix A. The bar on the bottom left defines the fraction of amino acid changes per site.

**Figure 4 microorganisms-10-00455-f004:**
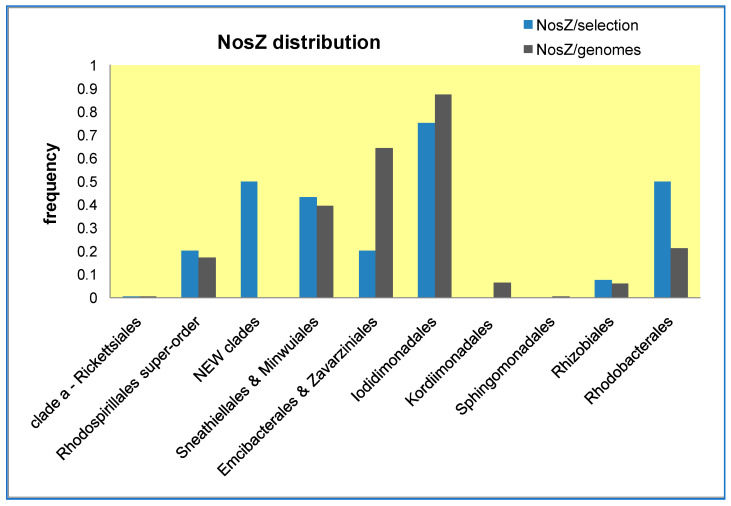
Distribution frequency of the gene for NosZ across Alphaproteobacterial lineages. The genomic distribution of the gene for NosZ has been estimated along Alphaproteobacteria lineages, ordered left to right by their branching order (Figure 1), using two different sets of normalization. The NosZ/genomes set (grey histograms) was calculated by dividing the number of proteins identified as TAT-dependent NosZ from a detailed search in the NCBI protein website (https://www.ncbi.nlm.nih.gov/protein/—accessed on 14 January 2022)—by the total number of deposited genomes available on the same date for each lineage. The NosZ/selection set (light blue histograms) was calculated from the presence of NosZ in a selection of genomes representative of each lineage, varying from 5 to 50 in proportion to the total number of genomes currently available for each lineage (cf. Figure 2b). The latter set was needed for estimating NosZ frequency in the new clades described here, for which there is very limited information in NCBI resources. Note that Alphaproteobacteria lineages that are not presented in the graph, for example Pelagibacterales, have no gene encoding recognizable NosZ proteins, as Rickettsiales and Holosporales.

**Figure 5 microorganisms-10-00455-f005:**
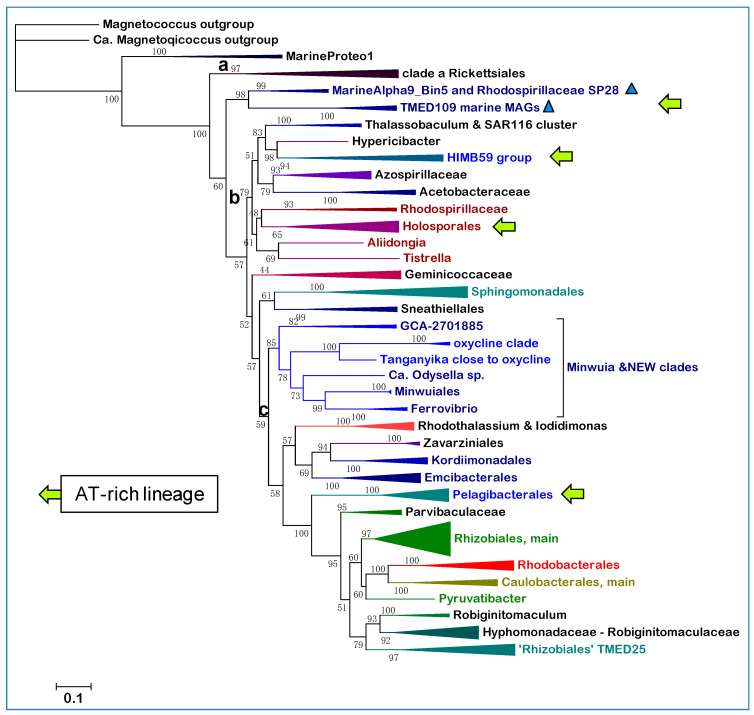
Phylogenetic tree of Alphaproteobacteria using a taxonomically broad set of NuoD sequences. The ML tree was reconstructed from an alignment of 142 proteins with a total length of 496 amino acid positions; its full set of annotated proteins is shown in Appendix A. The ML tree was obtained using the mixture model EX_EHO and four gamma categories as reported previously [23]. Trees with equivalent topologies were obtained using the WAG, LG and BLOSUM62 models with both ML and Bayesian inference. The arrows indicate the branches containing AT-rich lineages that are separate from each other as in heavily corrected trees [5,11,26]. The blue triangles indicate proteins maintaining vestigial Cys ligands of the Ni-Fe cofactor of hydrogenase ancestors of NuoD [22]. Rooting the tree on Gammaproteobacteria did not significantly change tree topology.

**Figure 6 microorganisms-10-00455-f006:**
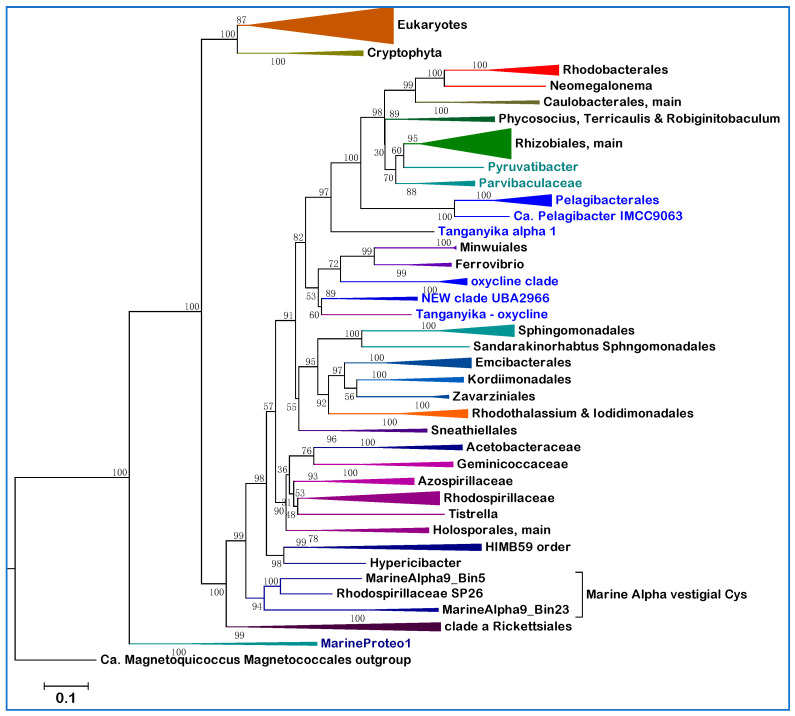
Phylogenetic tree of Alphaproteobacteria and mitochondria using an alignment of NuoD proteins with their mitochondrial homologs. The ML tree was reconstructed from an alignment of 145 proteins, 25 of which were Nad7 from Archaeplastida and other Protists, with a total length of 495 amino acid positions. The tree was obtained with IQ-Tree using the mixture model EX_EHO as in Figure 4. *Magnetococcus* outgroup (Figure 4) is cut off.

**Table 1 microorganisms-10-00455-t001:** Comparative taxonomy of the Rhodospirillales and Sneathiellales. The latest version of the NCBI taxonomy and GTDB websites were inspected to compare their diverse classification of Alphaproteobacteria in the simplest format possible. See Table 2 for other taxonomic divisions.

NCBI Order	Family	Genus	GTDB Taxonomic Division ^1^	Notes
Rhodospirillales	Acetobacteraceae	*Acetobacter*	o_Acetobacterales	
	Elioraeaceae	*Elioraea*	o_Acetobacterales	
	Azospirillaceae	*Azospirillum*	o_Azospirillales	
	Geminicoccaceae	*Geminicoccus*	o_Geminicoccales	
	Kiloniellaceae	*Kiloniella*	o_Kiloniellales	
	Reyranellaceae ^2^	*Reyranella*	o_Reyranellales	
	Rhodovibrionaceae ^2^	*Rhodovibrio*	f_Rhodovibrionaceae	
	Rhodospirillaceae	*Rhodospirillum*	o_Rhodospirillales	
		*Aliidongia*	o_Dongiales	
		*Defluviicoccus*	o_Defluviicoccales	
		*Elstera*	o_Elsterales	
		*Ferrovibrio*	o_Ferrovibrionales	SERIK ^3^
		*Magnetospirillum*	f_Magnetospirillaceae	
		*Tagaea*	f_ CACIAM-22H2	
		*Tistrella*	o_Tistrellales	
	Stellaceae	*Stella*	o_ATCC43930	
	Terasakiellaceae	*Terasakiella*	f_Terasakiellaceae	
	Thalassobaculaceae	*Thalassobaculum*	o_Thalassobaculales	
		*Nisea*	f_Niseaceae	
		*Oceanibaculum*	o_Oceanibaculales	
	Thalassospiraceae	*Thalassospira*	f_Thalassospiraceae	
		*Magnetospira*	f_Magnetospiraceae	
		*Magnetovibrio*	f_Magnetovibrionaceae	
	Zavarziniaceae	*Zavarzinia*	o_Zavarziniales	SERIK
	unclassified	various MAGs	o_UBA2966 and others	SERIK too
Minwuiales	Minwuiaceae	*Minwuia*	o_Minwuiales	SERIK
Sneathiellales	Sneathiellaceae	*Sneathiella*	o_Sneathiellales	SERIK
Emcibacterales	Emcibacteraceae	*Emcibacter*	o_Sphingomonadales, f_Emcibacteraceae	SERIK
Rhodothalassiales	Rhodothalassiaceae	*Rhodothalassium*	o_Sphingomonadales, f_Rhodothalassiaceae	SERIK
Iodidimonadales	Iodidimonadaceae	*Iodidimonas*	o_Sphingomonadales, f_Rhodothalassiaceae	SERIK
Kordiimonadales	Kordiimonadaceae	*Kordiiimonas*	o_Sphingomonadales, f_Kordiimonadaceae	SERIK

^1^ In GTDB taxonomy an order is denoted with the prefix o_ and a family with the prefix f_ [12]. ^2^ Rhodovibrionaceae and Reyranellaceae have been originally described as new families of the Rhodospirillales [13] but are currently affiliated to the Rhizobiales in NCBI taxonomy, for unknown reasons. ^3^ SERIK is the acronym formed by the initial letter of the orders Sneathiellales, Emcibacterales, Rhodothalassiales, Iodidimonadales and Kordiimonadales, as introduced here (see Figure 1).

## Data Availability

The alignments of NuoL and NuoD sequences shown here and the list of proteins used for Blast searches are available from the corresponding author upon request.

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
