# Peer review of "New Alphaproteobacteria Thrive in the Depths of the Ocean with Oxygen Gradient"

_microorganisms, 2022, doi:10.3390/microorganisms10020455_

Round 1

Reviewer 1 Report

The article by Cevallos and Esposti describes the phylogenomic analysis of the alpha-proteobacteria diversity, focusing on marine clades, and especially the functional differentiation of the different clades recovered based on the analysis of one reductase gene as well as concatenated alignment of universal genes. I believe the article is well-written, it requires minimal -if any- editorial editing, and offers a nice summary of the Alpha-proteobacteria, emphasizing the newly discovered clades and their functional and evolutionary distinctiveness. Hence, I believe there is enough novelty and useful information in the article. I have no major concerns with the manuscript; only a few minor comments for the authors to consider below.

I would suggest the authors to try to draw some more firm conclusions in the 1st and 3rd paragraphs of their Conclusions section as opposed to say -for instance- that different phylogenies should be examine etc., (that is, to say what is the best method of choice in their experience). Further, I urge the authors to taxonomically name/classify some of the new clades since they have studied their genomic discreteness and functional traits, toward increasing the (potential) impact of their paper.

Minor points:

For the title, please consider some alternatives that are more straightforward such as “ New Alphaproteobacteria clades associated with the depths...” (e.g., clades do not emerge from deep sea…)

Taxonomic names should be italics, including the phylum names since phylum is officially recognized as of last year (!).

Lines 84-85. Is it necessary to mention that the article is part of a special issue? Also, I think the first short paragraph of the Results section is enough so I would delete the lines 85-88 here that are somewhat redundant (or move the beginning of the Results here).

Line 251-252. Is photosynthesis apparatus prone to LGT? Traits with 100 or more genes are not thought to be much transferred horizontally, I believe. So, maybe it is the oppositive scenario; that is, the ancestor was photosynthetic and the trait was lost in some of the lineages? An alternative explanation to consider.

Figure 3. How selection was made? If proportional to the total genomes, I think showing total genomes is enough (that is, the second series is redundant or does not add much).

Lin 548-549. How similar is 16S gene phylogeny to that of nuo phylogeny? Would be interesting to mention (or even include as a supplementary figure), even if some MAGs probably do not have a 16S gene associated with them due to binning limitations. Keep in mind that 16S phylogeny is still the gold standard/reference for official prokaryotic taxonomy (e.g., it is not GTDB).

Author Response

We have highly appreciated the constructive comments by both Reviewers and dealt with them all as detail below. Our response to each specific point is preceded by a > symbol.

Reviewer 1.  The article by Cevallos and Esposti describes the phylogenomic analysis of the alpha-proteobacteria diversity, focusing on marine clades, and especially the functional differentiation of the different clades recovered based on the analysis of one reductase gene as well as concatenated alignment of universal genes. I believe the article is well-written, it requires minimal -if any- editorial editing, and offers a nice summary of the Alpha-proteobacteria, emphasizing the newly discovered clades and their functional and evolutionary distinctiveness. Hence, I believe there is enough novelty and useful information in the article. I have no major concerns with the manuscript; only a few minor comments for the authors to consider below.

>We thank the Reviewer for his positive comments.

I would suggest the authors to try to draw some more firm conclusions in the 1st and 3rd paragraphs of their Conclusions section as opposed to say -for instance- that different phylogenies should be examine etc., (that is, to say what is the best method of choice in their experience).

>Following the Reviewer’s suggestion, we have expanded the Conclusions section presenting some examples of our experience in classifying marine MAGs and the limits of current GTDB taxonomy. We have not examined enough taxa to make more general conclusions, as wished by Reviewer.

Further, I urge the authors to taxonomically name/classify some of the new clades since they have studied their genomic discreteness and functional traits, toward increasing the (potential) impact of their paper.

>Although we appreciate the suggestion to taxonomically classify the new clades we present in our paper, we believe that such a classification is still premature. We have expanded the phylogenetic analysis of, in particular, the oxycline clade, which is ‘better resolved phylogenetically (Fig. 3 - new) and functionally (Fig. 2B) than the other new clades’ as written in the expanded Conclusions section, p. 17 of the revised manuscript. We tentatively consider that the oxycline clade may represent a new family-level taxonomic unit of Alphaproteobacteria, distinct from either Rhodospirillales or Sneathiellales, as reported at p. 11 of the revised manuscript.  We hope to confirm this preliminary taxonomic evaluation once the MAGs reported in preprint of Ref [10] will be made public. In the future, we might be able to put together another paper dedicated to the full characterization of the new oxycline clade, especially if we find new members of the clade in metagenomes from Mexican oceanic studies conducted by colleagues we are in the process of collaborating with. Regarding the other clades, that overlapping GTDB order UBA2966 appears to have more phylogenetic breath than the oxycline clade, probably including some MAGs from lake Tanganyika as mentioned at p. 11 and 17 of the revised manuscript. The other clades fundamentally regard MAGs from lake Tanganyika for which there are too few relatives in current NCBI resources to define possible classifications – beyond the very tentative labels we have used in Fig. 2.

Minor points:

For the title, please consider some alternatives that are more straightforward such as “ New Alphaproteobacteria clades associated with the depths...” (e.g., clades do not emerge from deep sea…)  .

>Appreciating the suggestion, we have now modified the title.

Taxonomic names should be italics, including the phylum names since phylum is officially recognized as of last year (!).

>Point taken and corrections made.

Lines 84-85. Is it necessary to mention that the article is part of a special issue?

>Not really! Accordingly, we have re-structured the beginning and the end of the Introduction in the revised manuscript.

Also, I think the first short paragraph of the Results section is enough so I would delete the lines 85-88 here that are somewhat redundant (or move the beginning of the Results here).

>We agree and have deleted the noted text in the revised manuscript, following also general concerns of repetition raised by Reviewer 2.

Line 251-252. Is photosynthesis apparatus prone to LGT? Traits with 100 or more genes are not thought to be much transferred horizontally, I believe. So, maybe it is the oppositive scenario; that is, the ancestor was photosynthetic and the trait was lost in some of the lineages? An alternative explanation to consider.

>Well, we think that no alternative explanation is necessary, because the reality emerged from recent studies on Rhodobacterales, such as that of cited Ref [8], is that a genetic island carrying the numerous genes required for photosynthesis is frequently transferred among marine members of this order. A similar situation applies to photosynthetic marine members of the Rhizobiales and therefore our explanation is plausible also for Rhodothalassium and alpha Q-1. We have expanded the phrase citing this photosynthetic island prone to LGT at p. 7 of the revised manuscript.

Figure 3. How selection was made? If proportional to the total genomes, I think showing total genomes is enough (that is, the second series is redundant or does not add much).

>The selection of the taxa was made to provide a good representation of the genomic and physiological properties characteristic of each lineage, as succinctly written in the legend of previous Fig. 3, now Fig. 4 of the revised manuscript. We have now explained in the same legend, p. 12 of the revised manuscript, why this second method of evaluating the genomic frequency of NosZ is needed: there is very little information on the genomes of the new clades in current NCBI resources.

Lin 548-549. How similar is 16S gene phylogeny to that of nuo phylogeny? Would be interesting to mention (or even include as a supplementary figure), even if some MAGs probably do not have a 16S gene associated with them due to binning limitations. Keep in mind that 16S phylogeny is still the gold standard/reference for official prokaryotic taxonomy (e.g., it is not GTDB).

>Unfortunately, sequences specific for the 16S RNA gene are very difficult to find in most MAGs we have studied, and therefore the comparison mentioned by the Reviewer cannot be done. As an alternative approach, we have used the phylogeny of a nucleotide-reacting marker protein that is part of the 120 markers used for classifying bacteria in GTDB taxonomy, CTP synthase, as described in new sections and documented in Figure 3 of the revised manuscript. The results of this approach confirm and refine the phylogenetic data obtained with Nuo protein markers, as discussed at p. 11 of the revised manuscript.

Reviewer 2 Report

The recently increasing WGS and other NGS data have helped greatly to explore the untouched part of taxonomic classification. Recenlty, Hordt et al., 2020 (Forntiers in Microbiology), had attempted to improve the taxonomic classification of Alphaproteobacteria, however several issues and unresolved part of large group of taxonomic domain still prevails. This study has attempted to explore new clade within the Alphaproteobacteria from the depth of the ocean with oxygen gradient. Based on extensive phylogenetic analysis, new clades of Alphaproteobacteria  have been highlighted. The findings will obviously show positive impact on the existence status of the members of Alphaproteobacteria. However, besides phylogenetic and functional profiling, the authors should implement polyphasic approach which could adequately support their findings.

Overall, the manuscript has been written appropriately. Following comments could help the author to improve the manuscript:

  1. Abstract should contained their exact findings. There is no any clear information what the author has found from this studies. Please revise abstract.
  2. There are several repletion with the same information in several section of this manuscript. This should be corrected.
  3. A more elaboration on the significance of the explored new clade should be illustrated.

Author Response

We have highly appreciated the constructive comments by both Reviewers and dealt with them all as detail below. Our response to each specific point is preceded by a > symbol.

Reviewer 2.

The recently increasing WGS and other NGS data have helped greatly to explore the untouched part of taxonomic classification. Recenlty, Hordt et al., 2020 (Forntiers in Microbiology), had attempted to improve the taxonomic classification of Alphaproteobacteria, however several issues and unresolved part of large group of taxonomic domain still prevails. This study has attempted to explore new clade within the Alphaproteobacteria from the depth of the ocean with oxygen gradient. Based on extensive phylogenetic analysis, new clades of Alphaproteobacteria  have been highlighted. The findings will obviously show positive impact on the existence status of the members of Alphaproteobacteria.

>We thank the Reviewer for the succinct summary and perspectives of our paper.

However, besides phylogenetic and functional profiling, the authors should implement polyphasic approach which could adequately support their findings.

>As we explain in responding to the last comment of Reviewer 1, which goes in the same direction of undertaking complementary phylogenetic analysis with 16S RNA, the majority of the MAGs we are studying, including all those of the oxycline clade that we have analyzed in depth, do not have workable sequences of 16S RNA. Moreover, there is no member of the oxycline or the other new clades introduced in our paper that has been cultivated so far; therefore, classical polyphasic analysis of lipids, membrane quinones and growth features cannot be undertaken. However, we have shown functional data and genomic features such as %GC that have taxonomic value according to Hordt et al., 2020. In the revised data, we have added the complementary phylogenetic analysis obtained with the protein marker CTP synthase (new Figure 3) and several other data that provide a more defined picture of the possible taxonomic rank of, in particular, the oxycline clad, as detailed at p. 11 and in the Conclusions section of the revised manuscript.

Overall, the manuscript has been written appropriately. Following comments could help the author to improve the manuscript:

Abstract should contained their exact findings. There is no any clear information what the author has found from this studies. Please revise abstract.

>We have re-written the Abstract with a more fluid style and precise content in the revised manuscript.

There are several repletion with the same information in several section of this manuscript. This should be corrected.

>We have removed several instances of concept or data repetition that we encountered in the extensive revision of the whole paper, while we have added details and explanatory phrases to sharpen and clarify our scientific discourse.

A more elaboration on the significance of the explored new clade should be illustrated.

>We have expanded the Conclusions of the revised manuscript to duly meet this and similar comments of Reviewer 1.